# Certainty and Uncertainty in Tax Law: Do Opposites Attract?

**Alexander V. Demin**

Law Institute, Siberian Federal University, 660075 Krasnoyarsk, Russia; demin2002@mail.ru

**Abstract:** The principle of certainty of taxation is the dimension of a general requirement of certainty in the legal system. The purpose of this article is to argue the thesis that uncertainty in tax law is not always an absolute evil, sometimes it acts as a means of the most optimal (and in some cases the only possible) settlement of relations in the field of taxes. On the contrary, uncertainty and fragmentation in tax law are colossal problems subject to overcome by the efforts of scientists, legislators, judges, and practicing lawyers. Uncertainty in tax law is manifested in two ways: on the one hand, negatively—as a defect (omission) of the legislator and, on the other hand, positively—as a set of specific legal means and technologies that are purposefully used in lawmaking and law enforcement. In this context, relatively determined legal tools are an effective channel for transition from uncertainty to certainty in the field of taxation. A tendency towards increased use of relatively determined legal tools in lawmaking processes (for example, principles, evaluative concepts, judicial doctrines, standards of good faith and reasonableness, discretion, open-ended lists, recommendations, framework laws, silence of the law, presumptive taxation, analogy, etc.), and involving various actors (courts, law enforcement agencies and officials, international organizations, citizens, organizations and their associations) allow making tax laws more dynamic flexible, and adequate to changing realities of everyday life.

**Keywords:** tax law; taxation; certainty; uncertainty; tax rule; tax principle; relatively determined tools; vague concept

## 1. Introduction

Certainty in law is the cornerstone of the state governed by the rule of law.[1] The requirement of certainty must be respected both at the level of the legal system at large, and at the level of its individual area. The lack of certainty in law deprives it of its ability to perform its functions effectively, to form a stable legal order, and to direct (guide) people's behavior, in particular.

Tax relations require more precise regulations and control by the state. Therefore, certain requirements are imposed on certainty of taxation, and the principle of certainty of taxation is the ideological basis of all modern tax systems.[2] According to the correct observation of the reviewers of this work, certainty has several benefits. It lessens transaction costs to the taxpayer and the taxing authority, it facilitates

---

[1]   See Leoni (1972) (The author comes to the conclusion that the rule of law, in the classical sense of the expression, cannot be maintained without actually securing the certainty of the law, conceived as the possibility of long-run planning on the part of individuals in regard to their behavior in private life and business). See also Hayek (1960) (It is proclaimed that "the importance which the certainty of the law has for the smooth and efficient running of a free society can hardly be exaggerated. There is probably no single factor, which has contributed more to the prosperity of the West than the relative certainty of law which has prevailed here").

[2]   Recall one of the four Adam Smith's maxims: "The tax which each individual taxpayer is bound to pay ought to be certain, and not arbitrary. The time of payment, the manner of payment, the quantity to be paid, ought all to be clear and plain to the contributor, and to every other person" (Smith 1937, p. 371).

predictability and lets everybody plan his/her financial transactions, and it promotes faith in the system. The latter also makes tax collection palatable if not pleasant.

Issues of legal certainty are central to tax and legal science. This is due to a number of reasons. Firstly, tax law significantly restricts the rights of private actors and, primarily, property rights. Tax law includes a large number of coercive components, which requires preciseness in establishing the rights and duties of all relevant actors, as well as clearly established tax procedures. Secondly, tax law is excessively complex; it has significant economic content.[3]

Thirdly, tax reforms are carried out permanently, novels are regularly introduced into tax laws; therefore, tax norm setting is characterized by high dynamics, while instability is typical to tax law.[4]

Fourthly, the relationship between tax authorities and taxpayers is characterized by an acute conflict, caused by the mismatch (at times, antagonism) of the initial positions. The latter determines the difference in the identical tax rules interpretation by actors with opposite interests and needs.[5]

Unfortunately, the state of uncertainty, instability, inconsistency, and fragmentation of tax law becomes chronic. Measures taken by lawmakers cannot be called effective, since they are based on controversial methodological approaches. Continuous novelization of tax legislation, as a rule, does not reduce uncertainty, but only devalues the legislative process and produces tax disputes. In these conditions, paradigmatically new approaches are required, including deformation and decentralization of tax lawmaking processes with the involvement of a wide range of actors (courts, law enforcement agencies, international organizations, private actors, their unions, and associations) and the rejection of a one-sidedly negative attitude to the role of relatively determined legal tools in tax law.

Thus, the relevance of the research of paired categories "certainty" and "uncertainty" in tax law is obvious.

Tax law needs a thorough and forward-looking modernization to give it a new look, one that is adequate to the problems and challenges of the 21st century. It is required to create a single conceptual and methodological basis allowing to prospectively model, quickly identify, and effectively eliminate "zones of uncertainty" in tax law.

The first part of this study regards certainty as a general principle of law. It concludes that the requirement of certainty extends to all levels of legal impact—from the development of tax laws to their practical implementation of tax norms by addressees. Part I also analyzes why it is in the field of taxation that legal certainty is so important and necessary. Further, it discloses the content of the principle of certainty of taxation in its various aspects. Summing up, the first part draws conclusions about the role and significance of legal certainty for a successful impact of tax laws on social and economic interactions.

The second part analyzes the problems of uncertainty in tax law. In particular, it emphasizes that legal uncertainty should be considered in two aspects, namely: negatively—as a defect in tax legislation, and positively—as the use of relatively determined legal tools by the legislator in tax law. The latter includes legal tools with an open textured meaning (for example, legal principles, general standards of good faith and reasonableness, vague terms, open-ended lists, general anti-avoidance rules, silence of the law, discretion of fiscal authorities, presumptive taxation, legal analogy, etc.). The second part also considers the rules v. standards discussion in the context of "legal certainty v. legal uncertainty" dichotomy. It quotes "the principle of the taxpayer's rightness", applied in the tax law of Russia as an important guarantee for taxpayers.

---

[3]  See Paul (1997) (It is stressed that tax complexity is itself complex).
[4]  Gribnau (2013) (analyzing the positions of Joseph Raza, comes to the conclusion that an important principle connected to the ideal of legal certainty is that laws should be relatively stable; frequently changing laws hamper people in finding out what the law is at any given moment and they will not be sure the law has not been changed since the last time they learnt what it was).
[5]  See Givati (2009) ("Tax law is ambiguous in many cases. Different interpretations of the law are often possible, resulting in substantially different tax consequences. The inherent complexity of tax law and frequent changes in the law exacerbate this problem").

## 2. Principle of Certainty of Taxation Is the Fundamental Principle of Tax Law

*2.1. Certainty Is a General Requirement to Legal Norms*

Certainty is the most important trait (and the principle) of law as a universal regulator of social interactions.[6] The requirement of certainty follows from the very nature of the legal norm as a universal and equal scale (a measure) for all members of the society.

The European Court of Human Rights (ECHR) has formed stable assertions that legal certainty is one of the fundamental aspects of the rule of law.[7] According to the ECHR, every law must meet the legal certainty requirement ensuring that the rights of specific persons will be protected and, in any dispute resolution, the law enforcer's actions will be expectable and predictable and will not change from case to case.[8] Thus, interested persons with a reasonable degree of probability regarding the given circumstances can foresee the consequences of the current norms application and, in accordance with this, assess the consequences of the chosen behavior pattern.[9] The principle of certainty is aimed at maintaining reasonable regularity, stability, reliability, and predictability of the legal order, private confidence in the law, and the court.

In an academic discourse, the concept of "legal certainty" from the point of view of its content has a polysemantic and multilevel character, but is generally tied to the universal maxim of the rule of law.[10] In a narrow sense, the concept of legal certainty is limited to the norms of positive law, their specific legal and technical traits, i.e., the legal norms certainty is at issue.[11] More broadly, the requirement of certainty covers all levels of legal impact, including certainty, sustainability, and validity of judicial acts, as well as stability of the legal relationships that develop on their basis, so that interested persons with reasonable probability could foresee the consequences of legal norms application, and foresee the consequences of the choice of their behavior pattern.[12] In general, the level of certainty of the

---

[6] See Bertea (2008) ("Law and certainty are widely regarded as conceptually connected. By subjecting action to normative standards, the law limits the range of permissible conduct, thus reducing social contingency and superimposing an order on human interactions that would otherwise be less predictable and even potentially chaotic. So the very act of setting up a legal order reflects, among other things, a demand for certainty.... Certainty, thus, is not just one among several ideals by which legal practices can be assessed; it is rather a fundamental value of the legal domain"). See also Radbruch (1950).

[7] For the first time this, principle was mentioned in the ECHR Judgement in the case *Golder v. UK*, and concerned the inadmissibility of the broad interpretation of the European Convention on Human Rights (*Golder v. UK* (Application No. 4451/70), Judgment of the ECHR, 21 February 1975). The second Judgement, which mentions this principle, is the Judgement in the case "The Sunday Times v. The United Kingdom" (*The Sunday Times v. The United Kingdom* (Application No. 6538/74), Judgment of the ECHR, 26 April 1979). The third Judgement mentioning the principle of legal certainty banned retroactivity of court judgements (*Marckx v. Belgium* (Application No. 6833/74), Judgment of the ECHR, 13 June 1979).

[8] See Bergel (1988) ("The rule of law, a proposition destined to impose a certain behavior under social constraint, must be stated in a definite, concise, clear and precise manner").

[9] See Kelsen (1967), (It is commented that individuals who have to obey the law by behaving in a way that avoids sanctions, must understand the legal norms and therefore must ascertain their meaning).

[10] See, e.g., Bingham (2007) (the main component of the rule of law is that "the law must be accessible and so far as possible intelligible, clear and predictable"); Maxeiner (2008) ("Legal certainty is a central tenet of the rule of law as understood around the world"); Maxeiner (2006) ("The rule of law ... requires that rules of law be clear and consistent and that their application be sure and predictable. When that is true, law-abiding people can know what the law is and can orient their conduct on what it requires"); Gribnau, Hans, *Legal Certainty: A Matter of Principle*, p. 72 ("There are many conceptions of the rule of law, but the principle of legal certainty is generally recognized to be a key ingredient of it"); Morse (1999); Pagone (2009) ("Certainty in the law is fundamental to the rule of law, which holds that law should be clear, easily accessible, comprehensible, prospective rather than retrospective, and relatively stable").

[11] Gribnau, Hans, *Legal Certainty: A Matter of Principle*, p. 70 ("Rules play an important role in the human need for certainty. Rules offer structure, regularity, stability, reliability and predictability. Law as a body of rules aims at sharing these virtues. Legal rules enhance certainty of law").

[12] See Marmor (2018) (It is noted that vagueness in legal language can arise in many different contexts: in legislation or agency regulations, in constitutional documents, in judicial decisions, in private contracts, and wills, etc.) According to legal positivism, law is certain if the contents of general legal rules are knowable in advance, and public officials apply those rules uniformly and constantly to the effect that legal decisions can be said to be roughly predictable (Bertea, Stefano, *Op. cit.*, pp. 28–30). See also Raz (1979).

legal system can be judged by the degree of predictability of the results obtained while resolving legal disputes.[13]

Therefore, the concept of "legal certainty", apart from the requirements to the legal norms exposition, also includes the need for their uniform interpretation and application in practice. It is to the level of application of law that such a component of legal certainty, as legitimate expectations refers: every person acting lawfully and in good faith is entitled to believe that other persons (including the state and its representatives) will behave lawfully and in good faith.

Legal certainty has two aspects—external and internal. The external aspect is connected with formal enshrining legal norms in sources of law and their entry into legal force.[14] Legal norms are not just ideas, thoughts, intentions, and forms of public consciousness. They should be developed and adopted on the basis of the lawmaking procedure, properly promulgated, and enshrined in the official sources of law. The internal aspect concerns the content of normative texts: legal norms must be prescriptive statements that have a logical and semantic completeness. Moreover, special attention is paid to the legal language, which should be featured by accuracy, clarity, unambiguousness, emotional neutrality.[15] Thus, the language of law should be deprived of any expressiveness and literary pretentiousness.

From the position of legal certainty, legal rules must accurately fix the requirements that are imposed on people's behavior, on the one hand—must be laconic meeting the requirement of normative economy, and on the other hand—fully describe the scope of possible, proper, and forbidden conduct, as well as the consequences of offenses. It dates back to Roman lawyers who formulated the maxims: legem brevem esse oportet—the law should be concise; leges intellegi ab omnibus debent—laws should be clear to everyone; ubi jus incertum, ibi nullum—when the law is uncertain, it does not exist.

Despite its abstract nature, every the law should clearly outline life situations and give clear behavioral algorithms allowing of ensuring organization and predictability of social relations. The ECHR has repeatedly drawn attention to the fact that the law should make it possible to foresee the consequences of its application, responding to a standard requiring that every law be formulated with sufficient clarity that would allow a person by means of advice, if necessary, to foresee to a degree reasonable in the given circumstances, consequences, which may entail one or another of its actions (inaction).[16]

Preciseness, unambiguity, completeness, and consistency in exposing legal rules provide for a *uniform treatment*, i.e., extensive uniformity in understanding, interpreting and applying laws by their addressees. Ultimately, certainty of law is intended to guarantee a stable legal order being uniform for one and all.[17] Conversely, uncertainty of legal norms allows the possibility of unlimited and unjustified discretion in their interpretation and application, which inevitably provokes disagreements, disputes, legal conflicts. The lack of certainty turns law into its opposite, namely, into chaos and arbitrariness.[18]

---

[13] Notifying the difficulties in identifying the content of the concept "legal certainty", Vanessa Mak defines this concept as "the predictability of outcomes in legal disputes" (See Mak 2013).

[14] In the same sense, we can add such a requirement of legal certainty as non-retroactivity.

[15] In this context, Gustav Radbruch concludes that the legal language had the opportunity for its own independent development only after separation from other areas of culture and accordingly acquired a special aesthetic identity. The latter is based on the rejection of the numerous aesthetic values of the language. The language for law is cold, since it abandoned the language of senses. (See Radbruch 2004).

[16] See, e.g., *Ignatov v. Russian Federation* (Application No. 27193/02), Judgment of the ECHR, 24 May 2007; *Soloviev v. Russian Federation* (Application No. 2708/02), Judgment of the ECHR, 24 May 2007; *Melnikova v. Russian Federation* (Application No. 24552/02), Judgment of the ECHR, 21 June 2007; *Shukhardin v. Russian Federation* (Application No. 65734/01), Judgment of the ECHR, 28 June 2007; and many others.

[17] See Gribnau, Hans, *Legal Certainty: A Matter of Principle*, p. 69 ("People value legal certainty. Predictability of law protects those subject to the law from arbitrary state interference with their lives. Legal certainty enables people to plan their future").

[18] These positions are generally recognized in legal science. Thus, Friedrich August von Hayek has convincingly shown that clear and definite legal norms are immanently necessary for a free society based on the ideas of liberalism and a market economy (Hayek 1960). According to Ofer Raban, clear and determinate legal rules allow people to know where they stand (and where they should not stand) and therefore allow them to maximize their freedom (Raban 2010). Uncertainty of tax norms as Thorsten Jobs highlights, leads to the fact that citizens as taxpayers cannot, with the help of laws, foresee and

### 2.2. Why Is the Principle of Certainty So Important in Tax Law

All areas of law need certainty. However, this principle is especially relevant for tax law.[19]

The virtues of certainty for taxation include the following:

(1) Certainty of tax law increases the predictability and stability of taxation, allowing each taxpayer to rationally plan their financial transactions;

(2) Certainty reduces the transaction costs for the taxpayer and the tax authority in interpreting tax rules and finding criteria to assess their behavior as compliant; thereby, it contributes in every way to tax compliance;

(3) Certainty supports completeness and consistency in the regulation of tax relations, as well as the absence of contradictions in tax law;

(4) Certainty promotes faith in the tax system, making the tax collection palatable if not pleasant;

(5) Certainty of a tax rule guarantees its correct understanding, interpretation, and application.

The fact is that tax law is characterized by such a nature of regulation that implies limiting the constitutional right of ownership of private actors and distributing the burden of public expenditure among them taking into account the principles of universality and equality of taxation principles. This kind of regulation requires a clear definition of the limits of intervention in the field of fundamental rights and freedoms, including, first and foremost, ownership.[20] A high level of uncertainty in tax law is intolerable not only from the standpoint of requirements for legal technique, predictability of law enforcement, sustainability of economic activity, etc., but also from the point of view of the very essence of taxation that can be exercised only when it is sufficiently defined, regulated, stable, reliable, predictable, and non-contradictory. Being uncertain, tax law is unable to perform the function of allocating the burden of public expenditure on the basis of the principles of justice, equality and universality. Consequently, losing the quality of certainty, tax law turns into its direct opposite—to the mechanism of quasi-legal seizure of private property without due legal grounds and with vague goals.[21]

The style, logic, and the language of tax rules exposition play a major role in the formation of an effective tax system. The principle of the state governed by the rule of law requires the legislator to provide every taxpayer *ex ante* with credible data on the regulatory construction of the tax to properly fulfill the tax obligation. It is hardly possible to name another area of law, where an erroneously constructed phrase, a gap between a thought and its textual expression, an incorrectly or inappropriately used word can lead to such grave consequences as in the sphere of taxation.

The vagueness of the tax norm may lead to its arbitrary and discriminatory use by state bodies and officials in their relations with taxpayers, which is inconsistent with the principle of the rule of law, and thereby violates the principle of legal equality and the requirement of tax equality arising from it. Therefore, a tax provided for in defective norms cannot be considered legally established.

Despite the general legal requirement of certainty, the degree of the norms detail in different areas of law is not the same.[22] The more complex, conflicting, and publicly more significant the sphere (subject) of the law, the greater the share of mandatory rules, obligations, and prohibitions, procedural forms and coercive components, the more significant and deeper the involvement of the state in sectoral relations, the higher the requirements of certainty imposed on the norms of this branch of law. In this

---

calculate in advance whether they will be taxed and in what amount, and what exactly they are forbidden and what penalties are imposed in case of tax evasion (Jobs 2006).

[19] See Vasconcellos, R.P., *Op. cit.*, p. 4 ("Almost every person has to deal with taxes at some point in their lives. Hence, the need for less uncertainty in this field is stronger than in any other").

[20] See Pagone, Gaetano, *Op. cit.*, p. 887 ("Tax falls upon us in the ordinary course of our activities as a compulsory taking from us of something that we, by definition, have earned or owned. How and when that may happen should be clear, predictable and free from whim, caprice or chance").

[21] Cf.: Gribnau, Hans, *Legal Certainty: A Matter of Principle*, p. 93 ("Taxes constitute an interference with taxpayers' liberty. ... Established laws which offer certainty are a great security for the liberty of the taxpayers, for lack of certainty in tax legislation may leave them in the dark with regard to their fiscal rights and obligations").

[22] *Ibid*, p. 87 (Legislators can craft laws with different levels of specificity to guide human behaviour, incorporating detailed rules or more general standards in the laws they write).

context, tax relations require the most precise regulation and control by the state. Effective functioning of the tax system is impossible when the will of the state aimed at regulating tax relations is not strictly defined and equally understood by all addressees of tax norms.

Increased requirements for certainty in tax law are determined by a number of factors. This is, first of all, the public nature of tax law.[23] Taxes are the most important attribute of the state; their main purpose is to provide financial support for the implementation of state policies that is, for the normal functioning of society and the state. "Taxes are the life-blood of government".[24] Russian legislation expressly states that taxes are collected for the purpose of financing the activities of the State and (or) municipalities (Art. 8, para. 1 of the Tax Code of the Russian Federation). The public nature of tax law stipulates a special regime and methodology for regulating relations in the field of taxation. The latter is characterized by a significant specificity: (1) the priority of the common good while respecting a reasonable balance of public and private interests; (2) the authoritative and hierarchical nature of relations between tax authorities and taxpayers; (3) the prevalence of prohibitions and duties over rights and permissions; (4) the rule "taxes are not negotiable" applies; (5) active use of fiscal bodies' official interpretation, etc.

In addition, tax law is characterized by increased complexity[25] and high dynamics of changes introduced. This is due to the fact that tax norms are regulated mainly by economic relations, which are objectively inclined to self-development, diversity, and permanent transformations.[26] In modern conditions, the amplitude of such transformations is increasing, creating a threat of inconsistency between normative models, on the one hand, and actual situations on the other.[27]

The legislator's desire to ensure compliance of tax norms with the rapidly changing environment leads to continuous tax reforms.[28] Some of them are revolutionary, significantly affecting the social and economic life of the country.[29] This produces significant risks in terms of implementing the requirements of legal certainty, coherence, stability, and predictability of tax law.[30]

Increased attention to tax norms certainty is also stipulated by taxation being prone to high politicization and conflict. Why? The answer is simple. The sense of ownership is the oldest human instinct, and the instinct is innate, and not acquired in the process of social adaptation.[31] The desire to form, accumulate, and protect their property is rooted in every person at the subconscious level.[32] Some internal defense mechanisms force a person to resist encroachment on his property from any person, including the state. Therefore, every owner instinctively resists taxation, which in its essence is an individually non-refundable alienation of property.

Such a conflict of interests between a taxpayer and the state generates a multidirectional assessment and interpretation of tax norms and relevant facts. Such assessments are, as a rule, diametrically opposed. In these conditions, any "zone of uncertainty" in tax law—regardless of whether it was purposefully programmed by the legislator or formed as a defect of the law—is interpreted by

---

23　See Snape (2017) (noting that public purpose is a core element of identifying a charge as a tax in modern taxation).

24　*Bull v. United States*, 295 U.S. 247, 259 (1935).

25　Many authors stress out the increased complexity of tax law; See, e.g., Blank and Osofsky (2017); Raskolnikov (2013); Zelenak (2010); Givati (2009); Vasconcellos (2007); Pollack (1994); White (1990).

26　See Pouga Tinhaga (2013) (essay demonstrates that tax complexity is necessary for tax fairness through analysis of tax systems throughout the world).

27　See Evans et al. (2011) ("The complexity of tax law, like the complexity of the commercial world to which it applies, often seems to increase in an exponential fashion, placing ever more pressure on taxpayers, tax advisers and tax administrators").

28　Sometimes the term "legislative inflation" is used to describe the mass changes introduced into tax legislation (Eng 2002) and "constant flux of tax laws" (Gribnau, Hans, *Legal Certainty: A Matter of Principle*, p. 91). Gribnau ironically notices: "Consequently, tax laws nowadays look like throw-away articles and tax legislation is unstable" (*Ibid*).

29　See, e.g., Pearlman (1992) ("Taxpayers, as well as tax professionals both within and outside government, have been overwhelmed by the thousands of changes in the law").

30　See Givati, Y., *Op. cit.*, p. 146 (it is stated that the frequent changes in tax law are source of tax law uncertainty).

31　See Pipes (1999).

32　See Demin (2011b) (numerous observations suggest that along with the instincts of survival, domination, and procreation, proprietary aspirations determine the essence of *homo sapiens*. In our view, the sense of ownership is manifested in the form of a powerful instinct that is embedded in the human nature).

a person concerned in his favor.[33] Because of this, it is so important to ensure uniformity in the understanding and application of tax rules. The higher the level of certainty of tax laws is, the fewer the opportunities for their arbitrary interpretation and, therefore, for legal disputes and conflicts, resulting in an unproductive waste of time, efforts and energy are.[34]

Another aspect is often overlooked. Taxation always involves extrajudicial deprivation of property belonging to individuals.[35] The state here cannot rely solely on a sense of duty, tax morality, conformism, patriotism, every person's rational awareness of need for payment of taxes and other "internal" sources of incentive motivation.[36] Any legal obligation, including the payment of taxes, is provided by control and sanctions. Therefore, tax law is characterized by significant restrictions on human rights—especially private property, which requires certainty in establishing the rights, duties, and responsibilities of all participants in tax relations.

Tax law is largely a procedural area of the legal system. Tax law is the law of "legitimate violence" in the form of audits and the imposition of penalties, fines, and arrears. The possibility of applying state coercion measures is an indispensable guarantee for the lawful and conscientious implementation of tax norms.[37] At the same time, the more detailed the procedures for the application of state coercion in tax laws are, the more respected human rights are. That is why the ideas of formalism and literal interpretation are so popular with scholars and judges in terms of tax norms interpretation and application.

The immanently inherent in tax relations conflict connected with the unilateral movement of income from the taxpayer–owner to the state dictates increased requirements for detailed formalization of all stages of taxation, for their written fixation, strict adherence to practice. Without a carefully prescribed procedure, it is impossible, on the one hand, for taxpayers to faithfully fulfill the rights and duties provided for in tax laws, and on the other hand, to lawfully implement audits, tax control measures and apply tax sanctions.

Uncertainty in tax law produces significant transaction costs—material and temporal ones for all taxation participants. Taxpayers are forced to make extra efforts to find relevant responses and pay for expensive services of tax advisers.[38] In turn, tax authorities are forced to divert their limited tax administrator resources to prepare official interpretations and commentaries on tax laws. Of course, the greatest costs for all participants in tax relations are connected with the tax disputes resolution provoked by unspecified tax norms.

Therefore, in tax law, the requirement of certainty is expressed more categorically and persistently than in other areas of the legal system.

### 2.3. The Content of the Principle of Certainty of Taxation

The Tax Code of the Russian Federation treats certainty of taxation as an independent principle: "When taxes are established all elements of taxation must be defined. Acts of legislation concerning taxes and levies must be formulated in such a way that every person knows precisely which taxes

---

[33] See Christie (1964) ("[T]he slightest trace of vagueness will be exploited. Once vagueness has been found, one is free to choose the interpretation of which one approves").

[34] See Lawsky (2009) ("But tax law is complicated enough that it is unclear whether some tax positions are in fact wrong. Indeed, many tax positions that are eventually struck down by courts adhere to the letter of the statute").

[35] Gribnau, Hans, *Legal Certainty: A Matter of Principle*, pp. 80–81 ("Taxation is an interference with the right to enjoyment of property. In tax law, therefore, this certainty regards the "reach" of tax law and the inroads upon taxpayer's property right and economic freedom, i.e. certainty with regard to his tax burden").

[36] Of course, the significance of these factors is great for effective collection of taxes, and they should always be taken into account in assessments of tax compliance (See, e.g., Williamson 2017). I assert only that they cannot be fully relied upon when collecting taxes.

[37] See Lederman (2018).

[38] See Edmiston, K., Mudd, Sh., and Valev, N., *Op. cit.*, pp. 4–6 (problems under consideration connected with costs of complexity and uncertainty in the tax structure).

(levies) he must pay and when and according to what procedure he must pay them" (Art. 3, para. 6 of the Tax Code of the Russian Federation).

The content of the principle of certainty of taxation includes a number of interrelated imperatives, namely: preciseness and clarity of tax norms; understandability and accessibility of the tax rule for an "average" taxpayer; reasonable balance of abstract and concrete; completeness (no fragmentation); internal consistency and coherence (at least the absence of obvious contradictions) in the system of tax norms, where each norm must be agreed with other norms of Russian and international law.

Law scholars also often include other components in the principle of legal certainty.[39] For example, Lon Fuller adds such requirements as generality of law, the promulgation of laws, non-retroactivity, clarity, non-contradiction, and compliability to the concept of legal certainty.[40] In any case, a balanced account of all components of the principle of legal certainty is required, since an unreasonable "shift" towards one of them can give rise to a state of uncertainty in tax law.

### 2.3.1. Preciseness and Clarity

First of all, note that the tax laws should contain clear and understandable norms so that uncertainty in their understanding is not allowed and, consequently, there is no threat of their arbitrary interpretation and application. Formal certainty of tax norms implies their sufficient preciseness, which ensures their correct application.

The requirement of formal certainty, presupposing preciseness, and clarity of legislative prescriptions, is an integral element of the rule of law and acts in lawmaking and law enforcement activities as a necessary guarantee of effective protection of constitutional rights and freedoms.[41]

The requirement of preciseness and clarity means correspondence of spirit and letter of tax laws to each other. Logical and linguistic exposition of tax rules should clearly and unequivocally reflect the legislator's will (intention), he puts in when passing the law.[42] Of course, the legislator should, in every possible way, avoid ambiguity and indeterminacy of wordings.

Preciseness as a legal requirement means achieving the greatest correspondence between a thought (an idea) and the embodiment of this thought in the legislative formula.

Thus, the requirement of preciseness implies a mutual correspondence of the textual and semantic aspects of the tax norm that is, what it says, and what the legislator meant.

Lexical, stylistic, syntactic errors make it difficult to perceive tax laws, complicate their interpretation, and thereby produce tax disputes. Therefore, the addressees' adequate perception of tax norms and, consequently, the achievement of taxation purposes directly depends on how precisely the lawmaker expresses a thought (an idea) in a textual form.

An unclear tax law does not create a complete picture of legitimate behavior in a given situation, leading to misunderstandings, errors, contradictory interpretations, disputes, and conflicts.

### 2.3.2. Understandability and Accessibility

Tax norms should be concrete and understandable to everyone, targeted not at particular specialists, but at an "average" taxpayer. The tax laws should be simple to understand so that taxpayers can

---

[39] Takis Tridimas highlight that "legal certainty is by its nature diffuse, perhaps more so than any other general principle, and its precise content is difficult to pin down" (Tridimas 2006).

[40] See Fuller (1969).

[41] See, e.g., *Decision of the Constitutional Court of Russia of 12.07.2006 NO 267-O*. Bulletin of the Constitutional Court. 2006. NO 6. See also, Decision of the Constitutional Court of Russia of 06.04.2004 No. 7-P ("The adopted laws should be defined both in content and subject, purpose and scope of action, legal norms—formulated with sufficient preciseness, allowing the citizen to conform with them their behavior, both prohibited and permitted. Incomprehensible and contradictory legal regulation generates arbitrary law enforcement that violates these constitutional principles").

[42] See Bergel (1988) ("A legal terminology must be, above all, precise and exact. Certain concepts are identified by technical terms; common terms have generally acquired a proper legal meaning. But, each word should be a label through which one can identify with certainty one single concept. Polysemies must be avoided because they generate serious uncertainties and ambiguities").

anticipate the tax consequences of a transactions and activities, including knowing when, where and how the tax is to be paid.

The principle *"Leges intellegi ab omnibus debent"* operates. An incomprehensible law reduces its wordings' effectiveness, and in the worst case—does not allow it to be adequately implemented in practice.[43] Unfortunately, often not only taxpayers, but tax advisers as well cannot say exactly what the lawmaker meant in this or that tax norm.

The ECHR draws attention to understandability and accessibility of legislation as an important criterion for its compliance with universal values: the norm cannot be considered "the law" unless it is formulated with sufficient preciseness so that the citizen, independently or, if necessary, with professional assistance, could foresee, with a degree of probability, which can be considered reasonable in the given circumstances, consequences that may entail a particular action[44].

The tax law should be understandable and accessible, and be worded in such a way as to enable interested persons—by means of advice if necessary—to foresee, to the extent reasonable in concrete circumstances, the consequences that may result from certain acts.[45]

Of course, complexity and uncertainty are not equivalents. For example, a very voluminous law with numerous cross-references, many terms, and objects of regulation can be very difficult to understand, but at the same time, it is very detailed and clear from the point of view of legal language. In many cases, oversimplification in tax lawmaking can be detrimental. In particular, pressure to be succinct can lead to greater uncertainty.[46] Professors Joshua D. Blank and Leigh Osofsky observe that efforts to achieve the appearance of simplicity in areas of tax law can actually make matters more complex for taxpayers who are led astray by incomplete guidance.[47] Nevertheless, the excessive complication of tax law inevitably makes it difficult for the addressees of tax rules to understand it, provokes conflicting interpretations and disagreements, and, as a result, litigation that could have been avoided if the controversial tax rule were set out in a more understandable and generally accessible language.

### 2.3.3. The Balance of Abstract and Concrete

The balance of abstract and concrete in tax law means a reasonable combination of abstract and casuistic ways of tax norms exposition.[48]

The tax norm as an intellectually constructed formation is the result of the legislator's generalization of essential and typical qualities of tax relations. In this context, the tax norm is a typified model of a tax relationship.[49] The tax norm reflects "something common" in a variety of specific legal relations. For example, millions of taxpayers enter tax relations daily. Certain elements of these relationships are

---

[43] On the correct note of Roberto Vasconcellos "complexity leads to uncertainty and makes any legal rights and duties taxpayers may have unclear. . . . If legislation is to be applied to everyone, it should be made simple so people can understand it without having to spend more time and money" (Vasconcellos, R.P., *Op. cit.*, pp. 18, 33). The tax law, echoed Klaus Vogel, if it is so complex that even a professional is not able to understand its meaning, it not only cannot be fairly applied by government bodies, but contradicts the principle of the legislator's devotion to the idea of law as well (See Vogel 1994, p. 127).

[44] *Olsson v. Sweden* (Application No. 10465/83), Judgment of the ECHR, 24 March 1988.

[45] See *Liu and Liu v. Russian Federation* (Application No. 42086/05), Judgment of the ECHR, 6 December 2007.

[46] See Cauble (2019) (states that "lawmakers might simply refrain from issuing any guidance on various issues. This can lead to greater uncertainty").

[47] See Joshua D. Blank & Leigh Osofsky, Op. cit., pp. 192–94.

[48] In this context, it would be appropriate to cite the judgment of Herbert Hart: "in fact all systems, in different ways, compromise between two social needs: the need for certain rules which can, over great areas of conduct, safely be applied by private individuals to themselves without fresh official guidance or weighing up of social issues, and the need to leave open, for later settlements by an informed, official choice, issues which can only be properly appreciated and settled when they arise in a concrete case" Hart (1994).

[49] Gribnau, Hans, *Legal Certainty: A Matter of Principle*, pp. 71–72 ("This general law is opposed to any kind of individual command (rule of men). It is an abstract rule which does not mention particular cases or individually nominated persons, but is issued to apply to all cases and persons in the abstract. Law conceived as a general and abstract norm is attributed the intrinsic virtue of promoting certainty, equality, and liberty. With regard to legal certainty, the capacity of law to provide certainty depends its abstractness, which is a purely formal characteristic of law").

unique, inimitable, which is determined by the personality of a particular taxpayer, his tax culture, the amount of taxes and fees, the time and order of their payment, the presence of arrears, and other individual and often random factors. Moreover, at the same time, there is always something common in these legal relations that permits to adjust the diversity of this or that type of tax relations to normative models, where individual characteristics completely disappear, "dissolve" and are not taken into account.

The legal norm is a kind of averaged version between "too common" and "too concrete".[50] In this context, tax law refers to the most detailed areas of law, and the idea of the most precise and comprehensive tax regulation of "the whole lot" is widely supported by the tax community.

### 2.3.4. Completeness, Internal Consistency, Coherence

The most important imperatives of the principle of certainty of taxation are completeness and consistency in the regulation of tax relations, as well as the absence of contradictions in tax law. It is necessary to proceed from the fact that there cannot be anything superfluous in the text of the tax law: it is presumed that every word, every comma, every bracket, etc., carries its own semantic load.

Gaps and collisions of tax norms inevitably lead to legal uncertainty, encroaching on the basic quality of law—to be a coherent and balanced social regulator of people's behavior. Therefore, the tax law system must be a stable, non-contradictory, hierarchically organized set of tax rules interrelated and interacting with each other.[51] Tax rules should not contradict each other. Integrity and consistency, interdependence, hierarchy, internal consistency, coherence, and absence of fragmentation are necessary value characteristics of tax law, as well as any other area of law.

Summarizing the analysis of elements of legal certainty in tax law, it should be emphasized that the principle of certainty of taxation requires a weighed and balanced accounting of all imperatives making up its content. Unreasonable bias towards one of them can, in turn, generate a state of uncertainty.

### 2.4. The Role and Significance of the Principle of Certainty of Taxation

Significance of the principle of certainty of taxation is great and multifaceted. The quality of the tax system affects a variety of sociopolitical and macroeconomic indicators.

In particular, the country's investment climate directly depends on certainty of its tax legislation.[52] Investors need predictability in taxation, and this predictability can be ensured only with clear and precise tax legislation that does not allow arbitrary interpretations by either other taxpayers (as this distorts the competitive environment) or by tax authorities (since this leads to unplanned withdrawals of working assets and company profits).

The principle of certainty of taxation significance goes far beyond tax law own problems. This is a much broader spectrum of closely interrelated problems, including legal guarantees for the implementation of the constitutional principle of freedom of economic activity, further prospects for the development of the national economy, its stability, and attractiveness for investors.[53] In the context of global tax competition among states for investment and capital, creating attractive tax

---

50 Regarding the problem of the optimal abstractness of legal norms, R. David draws attention to the fact that the balance between the application of the norm in a specific situation and the general principles of law does not necessarily have to be the same in all areas of law: a great concretization is desirable in tax law, where they try to minimize the arbitrariness of the administration. On the contrary, a greater degree of generalization is necessary in some other areas where there is no need to strictly enforce rigid legal decisions (David and Brierley 1978).

51 Speaking of tax rules as consistent and coordinated with other legal rules, James Maxeiner acertains: "Legal rules are internally consistent. They are routinely coordinated with other legal rules of the same jurisdiction and with rules of other jurisdictions" (Maxeiner, James R., *Some Realism About Legal Certainty in the Globalization of the Rule of Law,* p. 40).

52 See Schwartzstein (1996) ("Uncertainty and instability of the tax legislation inevitably results in less investment, lower returns to investments and slower economic growth for the economy as a whole").

53 See Edmiston et al. (2005) (the present study indicate that complexity and uncertainty, in the sense of multiple tax rates, indeterminate language in the tax law, and inconsistent changes in the tax laws have a significant negative effect on inward foreign direct investment).

conditions, along with improving the infrastructure, increasing political system stability, economic factors, labor resources, etc., become one of the strategic and priority tasks of state policy.[54]

Taking into consideration the clash of conflicting interests of the state and taxpayers in tax law it is necessary to observe a reasonable balance between the interests of the former and the latter. This is the cornerstone of taxation.[55] The principle of certainty of taxation is a guarantee of public and private interest observance: on the one hand, its practical implementation restricts the discretion of tax authorities, which often grows into administrative arbitrariness,[56] on the other hand—prevents tax evasion by private actors. Uncertainty in tax law, on the contrary, can lead both to violations of the rights of taxpayers from the state and to tax violations and abuses—conscious or unintentional—on the part of taxpayers.

The principle of certainty of taxation is linked with the principle of equality. After all, an unclear tax norm can be applied or not applied to a taxpayer at the arbitrary discretion of the state, which violates the requirements of fairness and equality of taxation. The principle of certainty is stipulated by the constitutional principle of equality of all before the law and the court, since the latter can only be ensured by a uniform understanding and interpretation of the norm by law enforcement officials. Conversely, the indefinite content of legal norms allows for unlimited discretion in the process of law enforcement and inevitably leads to arbitrariness, and thus to violation of the principles of equality and the rule of law.

We must not forget that taxation has always been closely connected with a deep folk sense of justice; unbearable—in the eyes of the general public—taxation often provoked social tension, spontaneous protests, riots, and even revolutions.[57] Therefore, the social stability in society directly depends on the quality of tax law.

Thus, the lack of certainty in tax law is a double-edged sword. On the one hand, it allows the official, using his official position, to manipulate tax norms hiding behind a true or falsely understood common good, and on the other hand creates conditions for abuses by private actors, mass evasion from paying taxes. On the contrary, the creation of clear and transparent rules of the game reduces the danger of state arbitrariness, strengthens guarantees for citizens and organizations.

Consistent implementation of the principle of certainty of taxation ensures the clarity and concreteness of the rights and duties of the participants in tax relations, which allows them to plan their activities for the future without fear of accusations of violating the letter and the spirit of tax laws, and gives them confidence in the protection of their rights and freedoms.

Stabilization and unification of tax law, formalization of all stages of taxation, absence of fragmentation and gaps, consistent improvement of tax administration, stimulation of tax compliance, including not only the letter, but also the spirit of tax laws, ensure the achievement of a reasonable balance of public and private interests. Thus, predictability and stability of economic conditions are guaranteed. Ultimately, security of all subjects of tax law increases, tax law order is strengthened, structural and functional efficiency of the tax system is enhanced.

---

[54] A similar conclusion is made by many commentators; see, for example: Collier (2017); Pogorletskiy (2017); Slemrod (2009).

[55] See Vasconcellos, R.P., *Op. cit.*, pp. 3, 33 ("Judges in tax judicial review should not assume that the State rights are more important than the private rights. Both have to be outweighed in each case. . . . The idea of public interest can be misleading. It should not be seen as synonym of the state's interest, or the Treasury's interest. In fact, public interest can sometimes be found to be the same as the individual interest or the tax authority's interest. It should represent a fair balance of both interests at stake. The technical and impartial application of the law is part of the public interest. Therefore, there is no principle of superiority of the state's interest").

[56] See Logue (2007) ("It is suggested that tax administrators could use uncertainty as another tool in the tax compliance arsenal").

[57] See: Burg (2004).

## 3. Uncertainty in Tax Law: Causes and Consequences

*3.1. The Reasons for Uncertainty in Tax Law*

For many reasons, it is objectively impossible to achieve absolute legal certainty.[58] An absolutely definite tax law is a utopia, an unrealizable dream, an unattainable ideal, to which, of course, one must strive in every possible way.[59]

It seems obvious that the "uncertainty zones" are more or less present in every legal system[60]. Therefore, any attempts to create an impeccable tax legislation, which due to its preciseness and unambiguity would not require its creative interpretation by the law enforcer, are initially doomed to failure.[61] In general, we can state that uncertainty is an objective quality immanently inherent in all legal phenomena.

Uncertainty in tax law can be defined as lack of accuracy and clarity, understandability, and accessibility, completeness (i.e., the presence of gaps), a reasonable balance of abstract and concrete, stability, intra-industry, and inter-industry coherence in the system of tax norms.

Uncertainty can take various forms and develop as a result of not only of the legislator's activities, but of law enforcement entities' as well. That is, it can manifest itself not only in the texts of law, but also in interpretative and law enforcement acts, treaties, etc.[62]

Legal uncertainty can be determined by both objective and subjective factors. The latter relate to miscalculations, omissions, and deficiencies of the lawmaker while developing and modeling tax norms. The matter is the defects in legal technique stipulated by such reasons as the lack of the bill's concept reasoning or an erroneous failure to include tax norms that are still needed in the text of the law. In the end *errare humanum est*—the man is prone to make mistakes. Tax lawmaking being a product of people's consciously volitional activities is not free from mistakes, delusions, emotions, lobbying, voluntarism, etc.

Objective prerequisites of legal uncertainty are connected primarily with the semantics of legal language as a special kind of symbolic system,[63] subject to individual decoding and interpretation in each case of its application.[64] Exposition of tax norms in the textual form presupposes the possibility (and necessity) of their meaning and content interpretation. Polysemanticism, symbolism, and context are attributes of any complex legal text.[65] Therefore, any contact of the subject with a normative text is a creative and interpretive fact. As a result, it is possible to talk about certainty of tax norms only from

---

[58] Scott Baker and Alex Raskolnikov state that "few things are certain in life, and the legal system is not one of them. In a perfectly certain world, all laws would be clear, their application to the facts in each case would be unambiguous, as would be the facts themselves, all violations would be detected and punished, and sanctions would be fixed and known to everyone in advance. The reality, of course, is very different" (Baker and Raskolnikov 2017).

[59] In this context, Hans Gribnau writes: "the law never offers absolute legal certainty. It is only capable of providing limited certainty. ... Therefore, some uncertainty about the meaning of the law is inevitable. Absolute legal certainty, therefore, does not exist. Absolute legal certainty would mean that law and society at large would come to a halt. Therefore, some uncertainty about the meaning of the law is inevitable" (Gribnau, Hans, *Legal Certainty: A Matter of Principle*, p. 82). See also Kelsen (1992) (it is argued that certainty is a legal ideal, not a trait that systems of law necessarily possess).

[60] See Alexi (2005).

[61] See Baker and Raskolnikov (2017) ("Uncertainty is pervasive and persistent in any legal system. It comes in many forms, and it frequently reflects lawmakers' choices, whether deliberate or not").

[62] Yehonathan Givati names three types of possible manifestations of uncertainty in tax law: ambiguity concerning the precise meaning of statutory language, ambiguity concerning the application of the law to a specific factual situation, and ambiguity concerning the type of evidence sufficient to establish necessary specific factual situation, and ambiguity concerning the type of evidence sufficient to establish necessary facts (Givati, Y., *Op. cit.*, p. 144). See also Long and Swingen (1991).

[63] See Christie, George, *Op. cit.*, p. 911 (it is argued that vagueness is an inescapable aspect of our language and that vagueness is not always a hindrance to precise and effective communication; vagueness is sometimes an indispensable tool for the achievement of accuracy and precision in language, particularly in legal language; vagueness in legal language has also given our law a much needed flexibility).

[64] Cf.: Abreu, Alice and Greenstein, Richard, *The Rule of Law as a Law of Standards*, p. 92 ("The Internal Revenue Code does not 'read itself.' All applications of the Code require interpretation").

[65] See Pagone, Gaetano, *Op. cit.*, p. 887 ("Uncertainty may in part be an inevitable feature of language. Words are frequently capable of many meanings, some of which were not, or at least may not have been, intended when used in a particular context").

the position of relativity and conventionality, taking into account the textual nature and abstract form in which these norms are clothed.[66]

I assert that in relation to the content of tax norms, one can only talk about their greater or less certainty. The normative array of tax law can be represented in the form of a continuum of norms with two poles from "certainty" to "uncertainty".[67] Moreover, with respect to the tax norm, the "certain" (or "uncertain") characteristic will always be just some kind of conventionality, since every norm is formed using typification—that is, by abstract generalization of a number of selected traits within an ideal model. Therefore, it is methodologically incorrect, comparing two tax norms, to state that one of them is certain, and the second is not; it is more correct to state that the first norm is more certain than the second one. Thus, certainty is not an essential, but a qualitative characteristic of a particular legal norm, i.e., it is admissible to speak only of its certainty degree.[68]

In addition, because of their general nature, tax norms are always targeted at a certain "average" type of life situation and do not reflect the specifics of an infinite variety of tax situations.[69] Since the norm is an abstract generalization—that is, the result of the typification of some social relations, uncertainty lies in the very nature of law. Law is an artificially constructed representation of reality; therefore, it is only an approximate model of this reality. Owing to this, law reproduces images of real social relations only in the most general terms, but not verbatim.

Of course, the legislator by means of an abstract way of formulating tax norms is able to cover a wider range of homogeneous social relations, both existing at the time of issuing the statute, and those that may form in the future.

However, real life is richer than any regulatory system, so it is impossible (moreover, not even necessary) to try exhaustively to list all life situations that require a legal settlement.

It dates back to Roman lawyers who understood, *non possunt omnes articuli singillatim aut legibus comprehendi*—that it is impossible to embrace all individual cases by laws. It is impossible to take into account every nuance that can manifest itself in the course of the tax norm implementation. The dichotomy of concrete and abstract (in other words, certain and uncertain) objectively lies in the nature of tax law.

Thus, uncertainty zone availability in tax law is objectively conditioned by a typification method when formulating tax norms and a model form of their exposition.

However, not only an abstract, but also a casuistic way of norm formation in which actual circumstances are legalized by exhaustive enumeration or detailed specification of their main features, can also produce legal uncertainty.[70] Risk of uncertainty here is due to the fact that, resorting to minute detail, it is impossible to fully cover all the factors necessary for settlement, both existing at the time of passage of the law, and those that will arise in the future. Therefore, the casuistic way of tax norms exposition also provokes collisions and gaps appearance, which constitute the most dangerous types of uncertainty in tax law.

---

66　Cf.: Osofsky (2011) ("There are a number of sources of tax law uncertainty. Prevalent use of standards in the tax law may leave taxpayers unsure of the application of a standard to a particular set of facts. Tax rules often also create uncertainty. Incredibly complex tax rules may have small but gaping holes, leaving application of the rule uncertain").

67　See Joseph Raz's argument that rules and principles do not exist as distinctly separate standards, but they exist on a *spectrum of standards* of varying degrees of specificity (as quoted by McGarry (2017), p. 22). Mark van Hoeck, comparing general principles and precise rules, concludes: "In practice, there is no clear cut distinction between what is concrete enough to be considered to be a "rule" and what is general enough to be a "principle". It is a matter of gradation" (van Hoeck 2002, pp. 160–61).

68　According to H.L.A. Hart's correct remark, whatever norms (laws, precedents) we have chosen to communicate common patterns of behavior, in some specific situations they will always have what the scientist calls open texture of law. In this case, according to Hart, general legal terms have a core of certainty and a penumbra of doubt (Hart 1994, pp. 123, 128).

69　See Bertea, Stefano, *Op. cit.*, p. 32 (it is claimed that a law that can be qualified as certain provides uniform and entrenched generalizations that are relatively blind to the specificity of individual cases).

70　See Gribnau, Hans, *Legal Certainty: A Matter of Principle*, p. 87 ("Precision increases predictability. However, regulation through precise, specific rules does not always deliver optimal legal certainty").

The more detailed the tax norm, the more often problems with its operational adaptation to everyday life diversity appear.

Therefore, the level of detail and specification of tax norms—in the reduction of which one often sees the main way to combat uncertainty—has its limits. In any case, the requirements for accuracy, formalization, and detailed elaboration of tax law should have reasonable limits, beyond which they can turn into legal formalism, undermining the fairness and efficiency of the tax system.[71]

### 3.2. Relatively Determined Legal Tools in Tax Law

Today we live in a time of growing uncertainty in all areas of social interaction. There are no more familiar and centuries–established images of the world. Social time is accelerating. The changes multiply and grow.

This is the epoch of postmodernity and globalization—the era of rapid transformations and unpredictable consequences, when the ideas of diversity, instability, fragmentation, convergence, erasure of boundaries between established structures come to the fore. Moreover, the system of legal relations is no exception.

Radical complexity, diversity, instability, permanent development is inherent to a modern object of legal regulation. The latter, on the one hand, determines the avalanche-like growth of a legislative array, and on the other hand, the usual legislative mechanisms do not keep pace with accelerating changes in the legal regulation object. In the final analysis, zones of uncertainty in law expand, and attempts to cope with them through legislative solutions alone resemble patching holes, rather than solving problems.

Since social bonds and processes are infinitely diverse and dynamic, and they constantly evolve as the social life becomes more complex and globalized,[72] the legislator's ability to foresee the evolution of socio-economic phenomena and enshrine them in extremely precise rules is considerably limited.[73] One gap, eliminated by the lawmaker, is replaced by two, or even more ones.[74]

At present two strategic directions that of the unification and modernization of tax law are believed possible. The first direction is permanent updating tax legislation, i.e., transformation of the Tax Code into an expanded instruction with a simplified enactment of changes proposed. The second possible direction is active use of relatively determined legal tools in lawmaking and the transfer of some of the functions of norm formation to the level of law enforcement. Since numerous alterations, changes, and amendments regularly made to the Tax Code are not able to eliminate legal uncertainty and only devalue the legislative process,[75] the second way seems to be the most optimal one.

Active deformalization and decentralization of lawmaking, accompanied by pluralization of sources of law and involvement of a wide range of subjects in the lawmaking process: courts, law enforcement agencies, international organizations, private actors, their unions and associations, seem a promising direction of legal development. At the same time, the share and importance of relatively determined legal tools in the system of law increase. These are legal principles, common standards (for example, conscientiousness, reasonableness, economic validity), evaluative concepts,

---

[71] See Pashkevich (1982), p. 55 ("The lack of normative regulation or deficits in laws produce such negative effects as the disorganization of the public relations and the dangerous tyranny of bureaucracy. But there are also negative consequences in the case of overregulation, when the legal system has a lot of the redundant laws, over-detailed regulations, excessive formalization and bureaucratization, and so on").

[72] According to Friedrich Schauer's justified remark, the most precise rules are always potentially inaccurate, as a consequence of our imperfect knowledge of the world and our limited abilities to foresee the future (Schauer 1993).

[73] Even the most precise rule, Hans Gribnau emphasizes, may be uncertain when faced a situation that the legislator did not foresee at the time of laying down this rule (Gribnau 2007).

[74] See van Hoeck (2002) ("Moreover, social reality is constantly changing and the legal system has to adapt to this changed reality. It is actually impossible to do so by constantly changing legislation and amending each detail").

[75] Criticizing the legislator's hyperactivity, Walter Schwidetzky writes that as a result of excessive lawmaking "complexities are piled on top of complexities. Attempts to eliminate ambiguities rarely succeed; a law that resolves one ambiguity typically spawns many more. This whole process stems from the deluded belief that it is possible to have a perfect legal system. It is not. Humans are imperfect, and therefore so will be anything they create" (Schwidetzky 1996).

discretion, open-ended lists, framework (model) legislation, recommendations, silence of the law, use of judicial doctrines, presumptions and fictions, analogies, etc. A general legal tendency is obvious, and tax law is not an exception here.

Of course, the principle of certainty of taxation is an unconditional value and, as such, is not questioned. However, means and methods of its provision are paradigmatically changing. In considering global trends, the transition from detailed legislative regulation of "the whole lot" to a more flexible tax law and tax administration is observed everywhere, where some possibilities of establishing, specifying, and even developing precise content of tax norms are shifting from the legislator to the law enforcer.[76] With the help of open-ended legal tools, the state seems "to delegate" powers to continue the rule-making process up to the level of direct law realization (of course, where this is possible and admissible)[77]. In these conditions, all participants of legal interactions become partly "quasi-legislators".

Pluralization of the sources of law and decentralization of regulation (or outright deregulation) are worldwide tendencies, lawmakers are encouraged, whenever the practice is acceptable, to delegate the tailoring of some concrete aspects of legal norms to those to whom the norms are addressed. A wide range of subjects such as courts, law enforcement organs, international organizations, individuals, and their unions and associations are supposed to be involved in the lawmaking process. Very often, such concretization is executed with the help of relatively determined legal instruments. The risk here lies in blurring what is meant by lawful behavior. A lawmaker might adopt the following attitude toward the parties to taxation: "I know that you are clever and responsible people. I trust you. You understand how you must behave in order not to go too far. If it is difficult for you to understand, then begin a dialogue between the private actors and the authorities; involve consultants and the academic community. Uphold your position in court. I have set a norm, and it is your task to fill it with specific content." We all become lawmakers in part under these conditions.

It can be argued that the competent use of relatively determined legal tools is aimed at countering arbitrariness and entropy in law.[78] This is a specific transition from uncertainty to certainty in the tax law system.

It would seem that in the end we get the paradox: uncertainty versus uncertainty. However, this thesis is paradoxical only at first glance. Indeed, legal means with an open textured meaning include elements of uncertainty at the level of legislation, but at the same time, they allow to eliminate the state of uncertainty at the level of a specific situation. As a result of resolving specific legal issues, efficient and consensus-coordinated algorithms are developed, acting for participants in tax relations as authoritative models for making decisions and acquiring a de facto precedent character. Thus, discretionary arbitrariness in assessing and interpreting facts is limited to normative models that are developed by legal practice and which the law enforcer can (and has to) take into consideration in his practical activity. Ultimately, uniformity in the understanding and application of tax rules is ensured, and a due level of uniform treatment and predictability is achieved in the system of tax law.

Thus, sometimes legal uncertainty can be viewed not as a purely negative phenomenon subject to identification and eradication, but as a legal remedy consciously used by the lawgiver. In a positive sense, legal uncertainty is a technical device (tool) that the lawmaker uses and that allows to take into account the features and dynamics of the social relations development.[79]

---

[76] See, e.g., Gribnau, Hans, *Legal Certainty: A Matter of Principle*, p. 88 ("However, it is not always possible to formulate the subject matter that the law deals with in neatly tailored clear rules, or to mould it into precise concepts. Thus, the legislator may deliberately formulate rules (too) broadly, leaving it up to the courts or the tax authorities to tailor the rule more precisely").

[77] See Maxeiner, James R., *Legal Indeterminacy Made in America: U.S. Legal Methods and the Rule of Law*, pp. 522–23 ("[r]ules often intentionally grant those charged with applying them the authority to make value judgments. In such cases, rules provide only general limits, but no single correct answer").

[78] See Baker et al. (2004) ("The findings suggest that, within an efficiency framework, there are virtues to uncertainty that may cast doubt on the premise that law should always strive to be as predictable as possible").

[79] Cf.: Marmor, A., *Op. cit.*, p. 561 ("Vagueness in the law, as elsewhere, comes in different forms. Some of it is unavoidable, while other cases are optional and deliberately chosen by lawmakers").

The use of relatively determined legal tools in tax law makes it possible to envisage opportunities for individual legal regulation, provides law enforcers with an opportunity to choose the most expedient solutions, covers regulation of a wider range of homogeneous public relations, both existing at the time of the tax statute publication, and those that may form in the future. Everyday transformations of human bonds determine the need for adaptability and flexibility of legal (and tax) systems, and therefore an admissible degree of uncertainty in legislative provisions is the value of law, which helps to minimize the legislators' effort backlog from objective reality.

Many authors point out the possibility and appropriateness of using relatively determined legal tools (general principles, judicial doctrines, appraisal concepts, etc.) in tax law.[80]

For example, Ofer Raban argues that, in some cases, vague legal standards can provide greater certainty and predictability in many areas of the regulatory environment than bright-line rules.[81]

George Christie highlights that it is linguistic uncertainty that often allows law to exercise many of its social functions. In this context, the use of general terms with an open-textured meaning in law is not only inevitable, but rather necessary. Uncertainty, in his opinion, is sometimes an indispensable tool to achieve clarity and accuracy in legal language. In addition, legal language uncertainty endows all normative methods of social control with much-needed adaptability and flexibility. Christie believes that, without such flexibility, a person will have to choose between a complete lack of legal regulation and an impossible task of detailing what is possible and what is inadmissible.[82]

According to John Avery Jones, the desire for certainty in tax law through more and more detailed elaboration of legislation today is no longer working, as "detail and certainty do not necessarily go together". The desire to ensure the certainty of taxation through precise and detailed rules "results in more and more detail hoping to answer every question", which in turn produces extensive growth and complication of tax legislation, which is becoming increasingly complex and confusing. Jones calls this trend "tax rule madness" for sure; he sees a way out in the reorganization of tax law on the basis of more abstract general principles of taxation. In the end, he concludes that "we need less detailed legislation, construed in accordance with the principles, not a continuation of the plague of tax rule madness".[83]

Thorsten Jobs writes that the principle of certainty does not prohibit the legislator in the field of tax law to apply general rules, vague legal concepts, and references to other legal norms; tax laws binding a tax obligation with economic relations must bear entire diversity of economic realities; the principle of equal tax burden and tax justice can be realized only when tax authorities and courts can use vague legal concepts for specific cases, rather than adjust them to a rigid, final and casuistically formulated rule.[84]

Thus, uncertainty in tax law can be manifested in two ways. The first one is negative, that is, as a defect (omission). The second one is positive—as a specific technology of legal regulation consciously and purposefully applied in the process of tax lawmaking. In the first case, the situation of

---

[80] See, e.g., Weisbach (1999); Tittle (2006); Osofsky (2011).

[81] See Raban, Ofer, *Op. cit.*, pp. 177, 179 ("And in fact, clear rules are bound to produce less certainty and predictability than vague standards in many areas of the law. . . . As delineated above, capitalism and liberalism require the latter, not the former: what we want is a certain and predictable regulative environment (a predictable economic sphere, a predictable social sphere), not merely clear and determinate rules generating certain and predictable outcomes. And in fact, clear and determinate rules would often produce less predictable environments than vague legal standards").

[82] Christie, George, *Op. cit.*, pp. 885, 890 ("[i]t will be urged that it is precisely this vagueness in language which often permits the law to perform so many of its social functions. . . . The importance of the flexibility that vagueness gives to all normative methods of social control can scarcely be overestimated and is recognized by all. It allows man to exercise general control over his social development without committing himself in advance to any specific concrete course of action. Without such flexibility, man would have to choose between no regulation and the impossible task of minute specification of what is and what is not to be permitted. Moreover, if man tries to regulate too much in advance, he will be faced with the need to pervert his own language through the constant creation of vagueness in order to save himself from his own improvidence").

[83] Jones (1996) ("The real choice, I believe, is not between detailed rules that we have today and less detailed legislation, when detailed legislation wins on the ground of certainty; but between detailed rules and less detailed legislation interpreted in accordance with principles, where less detailed legislation wins on the ground of certainty because the use of principles provides predictability").

[84] See Jobs, Thorsten, *Op. cit.*, pp. 123–124.

uncertainty is an unconditional defect (imperfection) of tax legislation subject to elimination. In the second case, we are talking about relatively determined legal tools, which, although having an open textured meaning, allow reducing the overall level of uncertainty in the tax law system.

Various legal tools with an open textured meaning have relatively determined nature. Some of them are discussed below.

### 3.2.1. Principles of Taxation

The legal principles are characterized by the highest level of regulatory generalization (abstraction).[85] Due to its specificity—fundamentality, open-ended structure, multidimensional content, informative intenseness, axiological and ideological character, combination of direct and unwritten methods of exposition, etc.—they refer to the least detailed legal means.[86]

The principles of taxation have a complex structure. Unlike ordinary tax rules, they consist of a number of imperatives, some of which is of an unwritten nature.[87] Therefore, the content of the principle is not limited to once and for all given scope; it is multifaceted and changes with the development of tax and legal science and practice. The principle is polysemantic, while other tax rules have, for the most part, a very definite, unambiguous content.[88] The lawgiver can only outline the principle in general,[89] but further many subjects of tax law, primarily the courts reveal its content in the process of interpretation.

In tax law, the principles of universality and equality of taxation principles, the ability-to-pay principle, the principle of tax federalism, the principle of legal certainty and others are legalized and applied. At the same time, legal principles are applied in the most difficult cases, which cannot be resolved (settled, qualified) by using a "simple" tax rules.[90]

The principles of tax law cannot be viewed in isolation from each other; the operation of each of them is conditioned by the functioning of the entire system of principles as a whole. Organically complementing each other, they ensure the implementation of "horizontal" and "vertical" justice in tax relations. Regarding specific situations, they can dialectically "collide"[91] and then the law enforcer needs to make some effort to harmonize them with each other or to choose a principle that has priority in specific circumstances.[92]

---

[85]　Frans Vanistendael very figuratively describes the specifics of legal principles: "The use of legal principles in tax law is problematic, because it is like trying to explain astrophysics through the poetic beauty of a starstudded sky rather than through a mathematical model. . . . Floating in the galaxy of principles are a number of very general principles and concepts, which are seldom found in written texts of positive law, but are quite common in doctrine, jurisprudence and legal tradition and which are regularly used as instruments for the interpretation of law in general, and from time to time also in tax law" (Vanistendael 2014).

[86]　See, e.g., Atienza and Atienza and Manero (1998).

[87]　The official legalization of *the principles of tax law* can be different: some of them can be formulated as specialized rules, others are only mentioned in the sources of tax law, others are inductively derived from the context of a number of legal norms. Such legalization can be carried out by the legislator—in the process of issuing normative legal acts, by courts—within the framework of exercising their discretionary powers, by other participants in tax relations—by forming a stable law enforcement practice and its authorization by the state (See Demin 2011a).

[88]　Joseph Raz defines this difference as follows: "Rules prescribe relatively specific acts; principles prescribe highly unspecific actions" (Raz 1972, p. 823).

[89]　Mark van Hoeck notices that a legal principle will often be worded in vague terms (van Hoecke, Mark. *Op. cit.*, p. 160).

[90]　See Braithwaite, J., *Op. cit.*, pp. 3–4; Jones, J. A., *Op. cit.*, pp. 71–72.

[91]　Gribnau, Hans, *Legal Certainty: A Matter of Principle*, p. 79 ("[P]rinciples may collide, for example legal certainty and legal equality may point in different directions. Colliding principles make visible which values are really at stake on a deeper level. In the case of abuse or improper use of tax rules, for example, legal certainty, conceived as a principle, may constitute an argument not to change the law, and legal equality and the ability-to-pay principle may constitute an argument to change the law. Because principles do not dictate a decision or outcome but provide an argument pointing in a certain direction, the competing principles at hand ought to be balanced").

[92]　In resolving the disputable situation, the law-enforcing subject must choose and argue which of the principles in this case is most important; while all the conflicting principles retain their validity as acting regulators. Ronald Dworkin emphasizes this feature: "When principles intersect . . . one who must resolve the conflict has to take into account the relative weight of each. . . . Principles conflict and interact with one another, so that each principle that is relevant to a particular legal problem provides a reason arguing in favor of, but does not stipulate, a particular solution" (Dworkin 1977).

The main task is to make the principles of taxation not declarative but really effective, concentrate efforts on their use to resolve tax disputes, more fully reveal their *instrumental potential*, i.e., turn the principles into a daily attribute of practical jurisprudence.

### 3.2.2. The Vague (Evaluative) Terms and Concepts

The most obvious legal uncertainty is manifested when using tax norms with vague terms and concepts. It is assumed that the precise content of the vague term should be revealed in the course of its application in a specific situation. With that end in view, the legislator deliberately and purposefully does not detail the content of vague terms in tax laws, "delegating" such authorities to the addressees of tax rules. The latter receive considerable freedom in interpreting Vague terms and fill them with a varied content depending on the specific situation. Thus, uncertainty here is not a defect of lawmaking, but a special device of lawgiving.

Russian tax law applies a variety of evaluative concepts. Examples include such concepts as "income",[93] "justified expenses", "similar taxpayers", "assets, intended for everyday personal use", "information which is known to be false", "valid reason", "ancillary work (services)", "difficult personal or family circumstances", "insurmountable obstacle", "normal conditions", "regularly", "sufficient grounds", etc.

The vague (evaluative) concepts in the tax law are ambiguous and uncertain, their content has open character, and the law often does not contain guidance on how they should be understood.[94] Such specificity are forcing the addressee of the rule with the vague concept "to decode" individually and independently the meaning of the latter, putting into it his own understanding and case law. Open, i.e., incomplete structure of the vague concepts allows law-enforcers to supplement it with a new signs, and new meanings. Thus, the vague term can be metaphorically represented as a permanently constructed building, to which participants in legal relationships periodically add new "bricks"—additional structural elements.

If it is impossible for the precise concept to cover all the variety of phenomena regulated by law, alternatives to vague concepts are a gap or legislative inflation. Evaluative concepts allow of overcoming excessive formalism and inertia of tax law, make tax law more flexible and compliant, make it possible to take into account concrete situations specificity, ensure, on the one hand, compactness, and on the other hand—completeness of tax relations regulation.

The open-ended structure of evaluative terms relieves the legislator from the need to introduce permanent changes in legislation, contributing to its stabilization and predictability. Evaluative concepts are sort of "bridges" spanning tax law formalism and realities of everyday life.

Therefore, the strategic direction should not consist in refusing to use vague terms, but in finding *the optimal correlation* of detailed certainty and relative vagueness in the sources of tax law. It is required to optimize the total number of vague terms in the sources of tax law, to agree doctrinally their application methodology and to develop uniform criteria to evaluate and specify such concepts. The main thing here is the observance of equality and legal unification according to the principle "analogous solutions in analogous legal situations". Evaluation and specification of evaluative terms should be carried out within the framework and on the basis of the law, be grounded, supported by

---

[93] See about the vagueness and uncertainty of the concept "income": Abreu and Greenstein (2012) ("Instead of reflecting by its own terms tax law's defining values, it gives the IRS the flexibility to navigate the shoals of social opinion regarding income taxation, thereby both providing stability in the administration of the income tax and permitting the evolution of a concept of income that serves the important values in taxation").

[94] Earlier, we noted an open-ended structure of the principles of tax law, the content of which cannot be limited once and for all by the set limits. This feature is inherent in evaluative concepts, the content of which is not once and for all a given value, but constantly evolving. Ronald Dworkin points out the similarity of principles and evaluative concepts, in particular, noting that when an evaluated term is included in a rule wording, it makes the application of this rule "depend to some extent upon principles or policies lying beyond the rule, and in this way makes that rule itself more like a principle" (Dworkin, Ronald, *Op. cit.*, p. 28).

credible arguments, not based on a system of individual value landmarks, but rather on objective criteria and tests worked out empirically.

### 3.2.3. Open-Ended Lists

The open-ended lists of certain legal concepts (the so-called *catalogs*) envisaged by the Tax Code (e.g., interdependent persons; the circumstances which mitigate liability for the commission of a tax offence; the circumstances in which a person may not be found guilty of committing a tax offence; non-sale expenses; etc.) performs comparable functions as the vague concepts in the tax law. Like the Vague terms, they allow the tax authorities and the courts to expand (to supplement) such list with new elements.

### 3.2.4. Anti-Avoidance Rules

A special kind of relatively determined legal tools in the sphere of taxation is so-called general anti-avoidance rules (e.g., the doctrine of unjustified tax benefit, the substance over form doctrine, the sham transactions doctrine, the step transaction doctrine, the economic substance doctrine, the doctrine of an piercing the corporate veil, arm's length principle, the business purpose test, the substantive business activity, etc.).[95] The necessity for the address to them is caused by the fact that the rigid and unambiguous tax law is not able to prevent aggressive tax planning, which borders on the illegal tax evasion.[96]

Unfortunately, certainty of the tax law does not eliminate the possibility of its circumvention. Natural desire to avoid paying taxes encourages the taxpayer to tirelessly invent new ways of tax minimization, which often balance on the brink of legality. An individual may always find a loophole in the tax law that allows him to circumvent the law, but did not formally break it. Such play with rules creates boundless opportunities for the aggressive tax planning and for the designing the various kinds of the artificial tax avoidance schemes that distort the meaning and the purpose of the tax legislation. Tax abuses formally correspond to the letter of the law, but violate the spirit of the tax code.[97] With the help of precise and detailed tax rules alone, it is impossible to cope with this problem, as we will constantly come across a glaring discrepancy between legality and justice. In this aspect, the task of relatively determined legal tools, including general anti-avoidance rules, is to "wake up justice that has fallen asleep" in every tax law.[98]

Of course, the potential ex post application of one or more anti-abuse standards introduces uncertainty in tax planning practices, increases tax risks and makes the resolution of tax disputes unpredictable.[99] However, without resorting to them, one cannot effectively (if at all possible) combat

---

[95] Tax literature on GAAR is very extensive and multifaceted; See, e.g., Chand (2018); *GAARs—A Key Element of Tax Systems in the Post-BEPS World*. Michael Lang (ed.); Amsterdam—IBFD, 2016, 840p.; Rosenblatt (2014); Bank (2017); Feria (2017); Broe and Beckers (2017); Bykov and Frotscher (2016) and many others.

[96] Gribnau, Hans, *Legal Certainty: A Matter of Principle*, p. 88 ("Sometimes the legislator even deliberately formulates rules too broadly to put off taxpayers (chilling effect). Some anti-avoidance provisions to prevent tax evasion or abuse or undesirable use of tax legislation are a case in point. The result is uncertainty for taxpayers, leaving the courts large latitude to determine the application of the law").

[97] Sometimes tax literature and court decisions contain terms "substance of the law", "intent of the law", "the thing which the statute intended" and similar to them.

[98] Ilyin (1993, p. 255). See also Logue (2005). *Tax Law Uncertainty and the Role of Tax Insurance*, p. 374 ("[G]iven the existing incentives to under comply with the tax laws, maybe the deterrence value of a little legal uncertainty will at least help to even things out. Indeed, the only thing preventing some taxpayers from being more aggressive in their tax planning may be the residual level of uncertainty within the tax laws. And this might even be Congress's intenth lawmakers may have left some uncertainties in the tax laws with the hope in mind that taxpayers, seeking to avoid uncertainty, would err on the side of caution").

[99] See Givati, Y., *Op. cit.*, p. 146 (it is noted that the reliance on general anti-avoidance doctrines, such as the economic substance doctrine, also introduces significant uncertainty to tax law).

tax abuses, which often include several multi-stage transactions,[100] participants and tax jurisdictions, but do not reflect economic reality and produce tax consequences, not provided for by the tax law.[101]

In general, anti-avoidance rules prohibit the abusive behaviors in the sphere of taxes, but does not detail what constitutes such "the abuse". At best, the unspecified and indistinct criterions and reference points developed by judicial and law-enforcement practice are addressed taxpayers. Around the world, the opposition to such standards has been very strong;[102] however, today they are implemented everywhere as the general principles of tax legislation or as the judicial doctrines. Of course, the use of GAAR poses a real threat of the blurring of lawful behavior boundaries.[103] It is sometimes very difficult for a taxpayer to understand where the "red line" separates legal acceptability of behavior from abusive activity.[104] However, it should be noted that, in this case, the legislator has to choose not between bad and good, but between bad and worse.

### 3.2.5. Qualified Silence of the Law

Qualified silence of the law is a legislative technology consisting in conscious and deliberate unwillingness of the legislator to regulate this or that situation by means of legal rules.

Qualified silence as a method of legal technique is used in those cases when the lawmaker deliberately refuses from the normative regulation of the situation or regulates it in the most general form, delegating further settlement up to the level of the addressees of the law. In this case, the legislator does not say "no", "prohibited", "inadmissible", but recognizes that at the level of the law, the exhaustive regulation of the issue in question is unreasonable.[105] Therefore, he consciously leaves the question "open" in order to give persons concerned the opportunity of settling it in the way most convenient for them. There is a presumption that if the issue is not directly regulated by the tax law, everyone is entitled to exercise their rights and obligations in the manner most favorable for them.

In particular, mandatory forms of confirmation of many circumstances (for example, tax residency, the right to deduction, offset or refund of taxes, objections to the tax audit, various extracts from documents, notices, applications, requests, etc.) and the order of their submission to tax authorities are not officially established. This allows the taxpayer to independently determine the most convenient model of his behavior.

Another example: foreign organizations that have a number of economically autonomous subdivisions in the territory of the Russian Federation shall select a subdivision through whose place of registration with a tax authority, they will submit tax declarations, and pay tax in respect of the operations of all the foreign organization's economically autonomous subdivisions, in the territory of the Russian Federation taken as a whole. Foreign organizations shall give written notice of their choice to the tax authorities for the locations of their economically autonomous subdivisions in the territory of the Russian Federation (Art. 174, para. 7 of the Tax Code of the Russian Federation). Since the form of such notification and the term of its submission to the tax authority have not been legally established, the taxpayer is entitled to draw up a notification in an arbitrary form and send it to a tax authority at any time prior the deadline for filing a return and paying a tax.

---

[100] See Lawsky, Sarah B., *Probably? Understanding Tax Law's Uncertainty*, p. 1032 ("A transaction may adhere to every element laid out in the tax code but still violate the law").

[101] Blank, J. D., Staudt, N. C., *Op. cit.*, pp. 1647–49 ("While the latest forms of abusive corporate tax strategies are certainly more sophisticated than their predecessors, their basic objective—avoidance of corporate tax liability through an application of tax law that Congress never envisioned—remains the same. ... The mass marketing of these strategies by major accounting firms and other tax advisors in the late 1990s and early 2000s led to a corporate tax abuses, boom that commentators described in terms of these as an 'epidemic', a 'crisis', and a 'beast' that must be 'slayed'").

[102] See critical evaluations of the courts' application of general anti-avoidance rules: Hariton (2003); Shaviro and Weisbach (2002); Weisbach (2001); McMahon (2001).

[103] See Walpole and Evans (2011).

[104] Blank, J. D., Staudt, N. C., *Op. cit.*, pp. 1655–56 (It is noted that judicial outcomes over the past decade support the view that corporate tax abuse is an uncertain area of the law; judges may unevenly apply the judicial anti-abuse standards when determining whether or not a tax strategy represents tax abuse).

[105] It is important to distinguish between "silence of the law" and the gaps in the law; see Langenbucher (1998).

It is possible to draw up objections to a tax audit act in an arbitrary form, since there are no mandatory requirements for the formulation and content of such objections. An extract from the balance sheet, an extract from the sales ledger, an extract from the ledger of income and expenses and economic operations can also be filed with a tax authority in an arbitrary form.

Thus, with reference to "Silence of the Law", we are not talking about the drawbacks of lawmaking (as in the situation with gaps in law), but, on the contrary, about the deliberate expression of the will of the legislator, albeit expressed in a special way. In this case, the lawmaker intentionally transfers the right to resolve the issue to the addressee of the norm of law in order to take all the concrete facts and circumstances into account most effectively.

### 3.2.6. Discretion

Discretion is often manifested itself when the law gives the official the right to depart from the general pattern of behavior prescribed by the tax rule.[106] In this case, the official has to solve independently to realize to him this opportunity or not.[107]

In relation to the tax law, Ana Paula Dorado defines the term "administrative discretion" in its stricter sense as "the choice between two or among several different alternatives granted by law, and that choice implies a subjective assessment of the specific circumstances of the case which is not to be controlled by the courts". Thereby, in her opinion the discretion includes three elements: (1) that it is either explicitly or implicitly granted by law and, whereas in the former case there will be an express authorization by the law or statute in that direction, in the latter case that will stem from vagueness and indeterminacy; (2) that it requires a case-by-case assessment; (3) that the subjective assessment goes beyond interpretation, and that it must be exercised by the tax administration and therefore is not to be controlled by the courts, since, otherwise, a subjective assessment would be substituted for another subjective assessment.[108]

For example, the Russian legislators exhaustively lists the transfer pricing methods to be used in determining for taxation purposes income (profit, receipts) in transactions in which the parties are interdependent persons (Art. 105.7, para. 1 of the Tax Code of the Russian Federation). The comparable market price method shall be used on a priority basis for the purpose of determining the conformity of prices used in transactions to market prices; it is applied as a *general rule*. The use of the other transfer pricing methods envisaged in Art. 105.7, para. 1 of the Tax Code of the Russian Federation (the resale price method; the cost plus method; the comparable profits method; the profit split method) shall be permitted where the comparable market price method cannot be used (e.g., where there is no publicly available information on prices in comparable transactions involving identical (similar) goods, work and services) or where the use of that method would not enable a conclusion to be drawn on whether or not prices used in transactions conform to market prices for taxation purposes. The conclusion about the impossibility or the deficiency of the use of the comparable market price method, as well as the choice of one of the methods or combinations thereof is a discretionary power of the federal executive

---

[106] In legal science, the problematics of discretion is actively studied primarily in the context of judicial discretion and in the conditions of the total domination of sociological and logical-semantic law schools and traditions. The main discussion takes place not about admissibility of discretion per se in administrative practices—it actually is not challenged by modern experts, but around the limits and the scope (range) of the discretionary power. Specific approaches and definitions depend on the concept of law of specific authors. The most famous researcher of a discretion is Kenneth Culp Davis according to whom official "has discretion whenever the effective limits on his power leave him free to make a choice among possible courses of action or inaction" (See Davis 1976). For Israeli scientist Aharon Barak, discretion is "the powers given to the person who shall be entitled to the power to choose between two or more alternatives, when each of alternatives is legitimate" (Barak 1989). As Harold Pepinsky considers the discretion derives from accountability: "Accountability means having to answer for one's actions (or inaction). Accountability is thus synonymous with responsibility. Having to answer for one's actions makes sense only if one could have chosen to do otherwise". Eventually, he comes to the conclusion that "discretion amounts purely and simply to variance in decisions an observer is unable to explain" (Pepinsky 1984).

[107] See Zelenak (2012). See also Daly (2017) (states that decisions taken pursuant to tax authority's managerial discretion will only be disturbed by the courts where "exceptional circumstances" arise).

[108] See Dorado (2011, p. 30).

body in charge of control and supervision in the area of taxes and levies. As a criterion for the choice of "the best" transfer pricing method(s) the Tax Code indicates that the method to be used shall be that which, taking into account the actual circumstances and conditions of a controlled transaction, best enables a reasoned conclusion to be drawn as to whether or not the price used in a transaction conforms to market prices. In selecting the method to be used in determining for taxation purposes income (profit, receipts) in transactions in which the parties are interdependent persons, account must be taken of the completeness and reliability of source data and of the appropriateness of adjustments made for the purpose of rendering compared transactions comparable with the tested transaction (Art. 105.7, para. 3, 4, 6 of the Tax Code of the Russian Federation).

The reasons of use of a discretion coincide with the general factors that determine the use of the relatively determined legal tools in the tax law.[109] This, in particular, is the application of a teleological legal interpretation; the need of specification (and even more so—the need of individualization) of the abstract normative (regulatory) models generated by method of typification of common features of social interactions; the overcoming of excessive conservatism and formalism of statutory law, and as a consequence—the overcoming of inconsistencies and fragmentation in the law.[110]

One should agree with Gaetano Pagone, who claims that discretions in law, including tax law, may be necessary, but they should be structured, confined, reviewable, and above all predictable.[111] Ultimately, discretion plays an important role for the optimization of administrative impact in the concrete conditions of the tax administration, when it is required to react adequately, flexibly, and quickly to the specifics of individual cases, legal facts, or legal relation in general.[112]

### 3.2.7. Presumptive Taxation

In the area of taxes, there are the *special* relatively determined tools, which significantly expand the discretionary capacity of tax authorities. Examples include the imputed methods of calculation of tax liabilities, which are based not on the actual financial results (indicators), but on the conditional assumptions, presumptions, and analogies.[113]

In tax literature, such methods are often referred to as "presumptive taxation".[114] This method of taxation imputes income to businesses based on easily verifiable external factors, rather than relying on businesses to self-report their income.[115] As Victor Turonyi points out, presumptive taxation involves the use of indirect means to ascertain tax liability, which differ from the usual rules based on the

---

[109] See Pagone, Gaetano, *Op. cit.*, pp. 899–900 ("There are many reasons for discretions to be given in tax legislation notwithstanding the desirability of clarity, certainty and predictability One of them may be to have a tax outcome depend upon commercial, business or economic considerations that non-discretionary rules might not allow. . . . Another reason for discretions is that they are a response to what may be thought to be the 'social evil' of tax avoidance").

[110] As pointed out by Edward Morse, some discretion is unavoidable because of linguistic indeterminacy of the legislation: "Some forms of discretion . . . will remain a problem as long as human beings interpret language differently". In addition, he argues that discretion also arises from the practical impossibility of drafting rules to speak directly to every situation governed by law; in the areas of legal uncertainty discretion allows to fill in "gaps" between the rules (Morse 1999).

[111] Pagone, Gaetano, *Op. cit.*, p. 907.

[112] Analyzing the new forms of tax administration, Judith Freedman emphasizes: "An adherence to the rule of law does not mean that we must stick to the old cat-and-mouse game of detailed legislation, which often provides opportunities for taxpayers and their advisers to find ways of subverting that very legislation—the game of "creative compliance". The answer does not lie in rigid detailed legislation, literally interpreted; indeed, this is not the way most modern legal systems work, even in the tax area. It may be essential to leave some discretion in the hands of the tax authorities and the courts, but this must be bounded discretion" (Freedman 2012).

[113] See Ahmad and Stern (1991) ("The term presumptive taxation covers a number of procedures under which the 'desired' base for taxation (direct or indirect) is not itself measured but is inferred from some simple indicators which are more easily measured than the base itself").

[114] See Logue and Vettori (2011); Thomas (2013); Thuronyi (1998); Bird and Wallace (2005).

[115] Thomas, K. D., *Op. cit.*, p. 119 ("Under a true presumptive regime, the tax base is measured indirectly based on some readily observable characteristics of the taxpayer or the taxpayer's business").

taxpayer's accounts.[116] It is generally accepted that the presumptive taxation methods includes estimated assessments, standard assessments, and presumptive minimum taxes.[117]

In Russia, for example, tax authorities shall have the right to determine the amounts of taxes payable by taxpayers to the budget system of the Russian Federation using a calculation method on the basis of information, which is available to them concerning the taxpayer and data relating to other similar taxpayers if the taxpayer being inspected fundamental violates tax laws. In this case, the discretion is manifested above all in the search for and qualification of the third person as "similar taxpayer" because a calculation method of taxation is founded on the assumption that the other taxpayer, who is in good faith, engaged in the same kind of activity under similar economic conditions, has the tax base the amount of which is most likely assumed to be the same as the audited taxpayer. The basis for this discretion is the following presumption: the similar taxpayers have the similar tax bases. Certainly, two completely identical businesses cannot be found, they just do not exist in the nature. Therefore, presumptive methods of taxation are always only *relatively accurate* and *relatively reliable* that, however, does not call into question their legitimacy.[118]

The use of presumptive taxation is a forced measure, in demand in conditions of actual uncertainty, when it is difficult or impossible to obtain reliable knowledge about incomes and other elements of taxation. The main disadvantage of presumptive taxation is that here we deliberately sacrifice the accuracy (credibility) of calculating the tax, determining them with varying degrees of probability.

### 3.2.8. Intermediate Conclusions

Thus, relatively determined legal tools are a dimension of "positive uncertainty" in tax law.

The legislator should always pursue a preemptive tactic. In this context, relatively determined legal tools allow legal impact to cover not only existing realities, but also those that will arise in the future and, therefore, still unknown to the lawmaker at the time of passage of the law. This overcomes the contradiction between the high dynamics of social changes and the limited ability to foresee these changes. As a result, the fullest possible accounting of all probable situations related to taxation is ensured, and the ability of tax law to flexibly adapt to dynamically changing conditions and adequately respond to them is realized. Thus, the use of relatively determined legal tools is aimed at preventing incompleteness and uncertainty of tax law.

Of course, excessive enthusiasm for relatively determined legal tools can unjustifiably dilute the contours of tax obligations, increase instability, and provoke tax disputes. Therefore, the search for a reasonable balance between abstractness and concreteness, generalization and detailing, flexibility and rigidity, dynamism and stability in tax law is the most important task facing the tax community.[119]

---

[116] Thuronyi, Victor, *Op. cit.*, p. 401. ("The term 'presumptive' is used to indicate that there is a legal presumption that the taxpayer's income is no less than the amount resulting from application of the indirect method. As discussed below, this presumption may or may not be rebuttable. The concept covers a wide variety of alternative means of determining the tax base, ranging from methods of reconstructing income based on administrative practice, which can be rebutted by the taxpayer, to true minimum taxes with tax bases specified in legislation").

[117] See Taube and Tadesse (1996).

[118] See Logue, K. D. and Vettori, G.G., *Op. cit.*, pp. 103–4 ("A presumptive tax imposes a levy on one thing as a proxy for (or rough approximation of) another thing. ... Every tax system, of course, trades off accuracy for simplicity to some degree. And how much of a sacrifice in accuracy is required depends on the context. The term presumptive tax has traditionally been used to describe tax regimes in environments in which administrative/enforcement costs are unusually high and therefore accuracy of income measurement is unusually expensive. Such environments are often found in developing countries, where it is necessary to make unusually large sacrifices in income-measurement accuracy in order to be able to collect any taxes at all").

[119] See Pagone, Gaetano, *Op. cit.*, p. 906 ("Some uncertainty may be inevitable, but some is not. Certainty and uncertainty each come at a cost to the community, and our focus should be on what we gain and what we lose when we enact laws with deliberate uncertainties. We should look hard at who gains, and how much may be lost, from the uncertainty of the application of taxing laws and seriously question in whose interest uncertainty can be maintained").

### 3.3. The Principle of Resolving Doubts in Favor of Taxpayer

To reduce uncertainty in tax law, various legal means, and technologies are used. In Russia, the presumption that all unresolvable doubts, contradictions, and ambiguities in acts of tax and levy legislation is to be interpreted in favor of the taxpayer is of great importance.[120]

In the opinion of the Constitutional Court of the Russian Federation, the legal principle *in dubio contra fiscum* is the dimension of the constitutional principle of legally established taxes in tax law, by virtue of which the tax authorities can act only within the limits established by law enunciated in accordance with democratic procedures.[121]

This principle is termed "the presumption of taxpayers' rightness" and aimed at the protection of the interests of the latter. The fact-ground for this presumption is the existence of irremovable uncertainty in tax law, and a presumed fact is the taxpayer's right to interpret such uncertainty in his favor.[122]

Why does the law provide for such a "benefit" to the taxpayer? First, the private actor, opposing the state within tax relations, is an a priori "weak side" in subordinate tax relations. Therefore, protection of his rights and interests should dominate. With account for the unwritten principle of increased protection of the least protected counterparty, and in order to equalize the legal capabilities of the parties in tax disputes, the taxpayer is given priority over the state in interpreting irremovable doubts, ambiguities, and contradictions of tax law.[123]

Secondly, the presence of such defects is the fault of the rule maker, not the taxpayer. Therefore, it is the state as the guilty party that takes upon itself the burden of the negative consequences of all the shortcomings of the legislation. Since the lawmaker is obliged to formulate tax legislation in such a way that every person knows precisely which taxes (levies) he must pay, and when, and according to what procedure he must pay them, then it is the state that should be responsible for the non-fulfillment of this duty. Legal uncertainty caused by the legislator's insufficient work should be interpreted in favor of the taxpayer.[124] Thus, the presumption of the rightness of the taxpayer logically follows from the principle of certainty of taxation.[125]

Thirdly, the historical prerequisites for the formation of the presumption of the taxpayer's rightness can be found in the famous maxim of Roman law *in dubio pro reo* (or *in dubio pro tributario)*.

What do the terms "doubt", "contradiction" and "non-clarity" mean in the context of the presumption under consideration? Tax literature and judicial practice analysis shows that "doubt" is the impossibility of unambiguous interpretation of the tax rule content due to its uncertainty. "Contradiction" presupposes the presence of two or more tax rules of a mutually exclusive nature and equal force, as a result of which there arises uncertainty in the taxpayer's understanding of his rights and duties. "Uncertainty" is show up in the discrepancy between a semantic and textual component of

---

[120] Cf.: The Tax Ordinance Act of the Republic of Poland (Art. 2a) according to which "doubts about the content of the tax legislation, which cannot be eliminated shall be interpreted in favour of the taxpayer".

[121] *Decision of the Constitutional Court of Russia of 01.03.2010 NO. 430-O-O*. Bulletin of the Constitutional Court.2010. NO 4.

[122] Cf.: See Vasconcellos, R.P., *Op. cit.*, pp. 16, 33 ("Because the private property of individuals can be at stake when taxes are not paid or are paid less than what is due, it seems fair that, in case of doubt, the taxpayer should be considered exempted from payment. . . . If the obligation to pay tax is not made clear in the statute, the sanctions would represent a punishment on someone's intention".

[123] See Grossberg (2018) (discussing the doctrine of construing the tax code strictly against the government in the way that a contract is construed against the drafter).

[124] Cf.: *United States v. Merriam*, 263 U.S. 179, 188 (1923) ("[i]f the words [of a statute] are doubtful, the doubt must be resolved against the government and in favor of the taxpayer").

[125] See Juchniewicz and Stwoł (2017, p. 309) ("Application of the principle in dubio pro tributario in the interpretation of the tax law is to ensure implementation of the principle of certainty in tax legislation. This principle should perform the function of clarifying and simplifying legislation, which is important from the point of view of the general state of [t]ax law. This particularly applies to the situation when the position of regulation lead to conclusions that do not make sense, contradictory or ambiguous, then the best solution is a choice of interpretation of legal norms, which will be beneficial to the taxpayer").

the law fragment, objectively impeding the precise interpretation of the actual will of the lawmaker, expressed in such a fragment.

Take notice, that in this case it is a matter not of any legislative defects, but only of collisions of irremovable character.[126] According to the elucidation of the Constitutional Court of the Russian Federation, certain legal and technical inaccuracies permitted by the lawmaker when formulating the tax rule, while making it difficult to understand the true meaning of the law, are not grounds for concluding that such a norm is indeterminate, vague, not containing clear standards, and respectively, not meeting the constitutional principles of taxation[127].

Uncertainty that is not irremovable, in the Court's view, should be overcome by systematic interpretation, with regard for hierarchical construction of rules in the legal system, which assumes that lower-level rules should be interpreted in accordance with higher-level ones and with account taken of general objectives of the relevant law passage.[128]

Thus, in order to apply the "presumption of the taxpayer's rightness" it is not enough to establish the presence of ambiguities or contradictions in the tax law, it is also necessary to ground their irremovable nature. The latter means that it is impossible to eliminate legal uncertainty by interpretation, when the only possible means of its eliminating is lawmaking.

Considering tax disputes arisen due to different interpretations by tax authorities and taxpayers of tax laws, courts should assess the certainty of relevant tax rules. Moreover, the presumption of the taxpayers' rightness can only be applied as an extreme measure, when any other legal means to resolve a tax dispute has already been exhausted. At the same time, only those doubts that cannot be eliminated can be recognized unremovable, irrespective of the use of all known methods of interpreting the law: grammatical, logical, historical, and other methods, comparative legal analysis of this rule, and related tax norms, as well as by means of direct application of tax law fundamental principles.[129] In any case, uncertainty in tax law is not recognized as unremovable if there are legitimate legal technologies for its resolution.

For example, contradictions between the legal norms of the same legal force are unremovable. Gaps in tax law, which are unremovable to overcome by analogy, are of unremovable nature.[130] Lack of uniformity on the tax norm interpretation in judicial practice and official explanations of fiscal authorities can serve as a confirmation of non-removability. It should be emphasized that for the court issuing incompatible elucidations of state bodies on a contentious issue is not an unconditional proof of non-removability of doubts, contradictions, and ambiguities in tax legislation.

Some common, universal criteria and tests for applying the presumption of the taxpayer's rightness in practice have not yet been formed. In each specific case, the court must, in its internal conviction, fully and comprehensively examine and evaluate the totality of the evidence submitted by the parties to a case.

---

[126]  See Vasconcellos, R.P., *Op. cit.*, p. 17 ("The real problem lies in determining whether the law (legislation and relevant precedents) is uncertain or not at some point").

[127]  *Decision of the Constitutional Court of Russia of 28.03.2000 NO 5-P*. Bulletin of the Constitutional Court. 2000. NO 4.

[128]  *Decision of the Constitutional Court of Russia of 13.03.2008 NO 5-P*. Bulletin of the Constitutional Court. 2008. No 3.

[129]  For example, the contradiction of rules contained in sources of law of different legal force is easily removed by applying the *lex superior derogat legi inferiori* principle, known since the Roman law. The collision between the general and special norms is resolved on the basis of the principle *lex specialis derogat generali*. If there is a contradiction between the norm contained in the Tax Code and the norms of other (uncodified) tax laws, the priority of the codified act is valid. In addition, Russian tax law has a rule that "Institutions, concepts and terms contained in civil, family and other areas of legislation of the Russian Federation which are used in Tax Code shall have the same meaning as they have in those areas of legislation, unless otherwise stipulated by Tax Code" (Art. 11, para. 1 of the Tax Code of the Russian Federation).

[130]  This axiom relates to the sphere of taxation (subjects and elements of taxation), as well as the qualification of an act as a tax offense and the establishment of tax sanctions. See Langenbucher, K.C., *Argument by Analogy in European Law.* Cambridge Law Journal, Vol. 57, No. 3, 1998, p. 486 (" . . . no revenue authority may impose taxes not explicitly permitted by the law and no judge may sanction taxation outside the scope of a statute. Where only the legislature may legitimately act, the judiciary may not extend its powers by means of an analogy").

### 4. Conclusions

Thus, the principle of certainty of taxation includes a number of formal and rich in content requirements, namely: preciseness, clarity, understandability, and accessibility for a general understanding of tax norms, a reasonable balance of abstract and concrete, completeness (the absence of fragmentation), stability of tax legislation, logical and systemic consistency of tax norms, i.e., coherence (at least—the absence of obvious contradictions), where every new norm must be harmonized with the norms of the national and international law.

Uncertainty in tax law is manifested in two ways: on the one hand, negatively—as an omission of the legislator and, on the other hand, positively—as a combination of specific legal means and technologies that are purposefully applied in lawmaking and law enforcement. In the first case, it is a matter of defects in the tax law, subject to elimination in the process of lawmaking; in the second—the point is special methods of legal techniques, which, although are relatively determined in nature, allow reducing the overall level of uncertainty in tax law.

It seems obvious that relative certainty is always better than total uncertainty. Today we are witnessing the formation of new trends in the system of legal regulation, both at the sectoral level and at the level of the legal system at large. The matter is a peculiar quasi-delegation of legislative power-ups to the level of law enforcement.

In this context, relatively determined legal tools are an effective channel for the transition from uncertainty to certainty in the field of taxation. The paradoxical dialectic of modern law perception is that the tendency to expand the use of relatively determined legal tools in the processes of lawmaking, and the involvement of mass actors in it, on the one hand, increases the overall level of legal uncertainty, but on the other hand, it allows tax law to be more flexible and adequate to the rapidly changing realities of modern life.

The use of relatively determined legal tools is a kind of "invitation" to rulemaking. It is here that the creative abilities of a lawyer, his ability to think unconventionally, to make non-standard decisions, to operate with interdisciplinary categories, including those outside the legal system are revealed.

The legislator resorts to relatively determined legal tools to, firstly, provide tax relations participants with legitimate behavior algorithms in situations of uncertainty in tax law and, secondly, to give tax law greater flexibility and elasticity to promptly respond to environmental evolution. Such means contain elements of uncertainty at the level of legislation; but at the same time, they allow of eliminating uncertainty at the level of a specific situation. Ultimately, unification in the understanding and application of tax rules is achieved.

The tendency to expanded use of relatively determined legal tools (principles, evaluative concepts, judicial doctrines, standards of good faith and reasonableness, discretion, open-ended lists, recommendatory acts, framework laws, silence of the law, presumptive taxation, analogy, etc.) in lawmaking processes, and the involvement of various subjects (courts, law enforcement agencies and officials, international organizations, interstate integration bodies, private actors and their associations) in it allows making tax law more dynamic, "mobile" and adequate to the changing realities of everyday life.

The main task is to find the optimal balance between rigidity and flexibility of tax norms, ensuring, on the one hand, predictability, regularity, and uniformity of tax law, and on the other hand, its dynamic development, viability, and adaptability in the era of globalization and accelerating environmental transformations.

**Funding:** This research received no external funding.

**Conflicts of Interest:** The author declares no conflict of interest.

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
