# Peer review of "Certainty and Uncertainty in Tax Law: Do Opposites Attract?"

_laws_

Round 1
Reviewer 1 Report
While I liked this paper, there is room for improvement but it has potential. My biggest criticism is that when I began the paper, I expected it to tell me that certainty has several benefits. It lessens transaction costs to the taxpayer and the taxing authority, it facilitates predictability and lets me plan my financial transactions, and it promotes faith in the system. The latter also makes tax collection palatable if not pleasant.
The basic ideas outlined above appear, not in a cohesive form in the introduction but seem scattered throughout the paper. Not until the second page do the authors mention that certainty contributes to “a stable legal order and to direct (guide) people’s behavior.” Not until page four is the reader informed that “certainty is aimed at maintaining reasonable regularity, stability, reliability and predictability.” It is not until page nine that the reader is told that “[u]ncertainty in tax law produces significant transaction costs.” Finally, on page twelve the reader is informed “[t]he most important imperatives of the principle of certainty of taxation are completeness and consistency in the regulation of tax relations, as well as the absence of contradictions in tax law.” While I am inclined to quibble about the characterization, a reasonable question to ask is that if these, completeness and consistency, are the most important imperatives of the principle of certainty, why aren’t these imperatives introduced at the outset?
All of this is to set forth my first constructive criticism. The reader should get a sense of where the paper is heading in the beginning. What is needed is simple declarative sentence to the effect that certainty has a series of benefits – then explicitly name the benefits – which the authors will then examine in order. The paper can then proceed in a linear and logical way.
Beyond the general lack of organization of the paper, I have a number of comments.
- On page two, the paper attributes uncertainty to the structural tendency to complexity. The difficulty with this formulation is that complexity is not necessarily the equivalent of uncertainty? For example, Schoederinger’s wave equation is complex but it is not uncertain.
- I liked the insight mentioned beginning on page five that “[l]egal certainty has two aspects – _external and internal.” This idea could be expanded somewhat and would make the paper stronger.
- Footnote 16 on page five contained a reference to a law review article written by the late U.S Supreme Court Justice Antonin Scalia without seeming to acknowledge his stature. Justice Scalia was the ultimate textualist. I’m not sure that the citation supported the thesis or did it intend to represent the extremes of certainty. I’d like to know what the authors thought of this reference.
- Page thirteen brought up the position of the Constitutional Court of the Russian Federation. I wasn’t sure why the reader should care about this court or its views. A better introduction to the idea of why the court’s view matters would be useful.
- Page fifteen introduced the idea of corruption. This was a jarring transition. The foray into corruption seemed misplaced. While I have little doubt that corruption is omnipresent in varying degrees, the tie-in of certainty and corruption was itself not so certain.
- An area that could have been better developed is the relationship of tax-avoidance and certainty that is discussed briefly on page twenty-five.
My bottom-line assessment is that this is a paper worth publishing. What it needs is better organization, linear progression, clear separation of ideas and how they interrelate, and a better development of some themes. As to the latter my preference would be on tax-avoidance if a practical application of the concept of certainty is desired or on Scalia’s strict textualism if a more theoretical discussion is the aim.
Reviewer 2 Report
There is no clear definition of the purpose of the article in the abstract. Moreover, the abstract should be shorter, some of the information can be placed into the Introduction.
